# Margin based Self-Supervised Neural Architecture Search

## Abstract

Neural Architecture Search (NAS) has been used recently to achieve improved performance in various tasks and most prominently in image classification. Yet, most search strategies rely on large labeled datasets, which limit their usage in the case where only a smaller fraction of the data is annotated. Self-supervised learning has shown great promise in training neural networks using unlabeled data. In this work, we propose a self-supervised neural architecture search (SSNAS) that allows finding novel network models without the need for labeled data. We show that such a search leads to comparable results to supervised training with a "fully labeled" NAS. While such a result has been shown in concurrent works, the uniqueness of this work is that we also show that such a search can also improve the performance of self-supervised learning. We show that using the learned architectures for self-supervised representation learning leads to improved performance. Thus, SSL can both improve NAS and be improved by it. Specifically, due to the common case of resource constrains, we exhibit the advantage of our approach when the number of labels in the search is relatively small.

## 1 Introduction

Recently there has been an increasing interest in Neural Architecture Search (NAS). NAS algorithms emerge as a powerful platform for discovering superior network architectures, which may save time and effort of human-experts. The discovered architectures have achieved state-of-the-art results in several tasks such as image classification Xie et al. (2020); Touvron et al. (2019) and object detection Wang et al. (2020).

The existing body of research on NAS investigated several common search strategies. Reinforcement learning Zoph & V. Le (2017) and evolutionary algorithms Real et al. (2017; 2018) were proven to be successful but required many computational resources. Various recent methods managed to reduce search time significantly. For example, Liu et al. (2019b) suggested relaxing the search space to be continuous. This allowed them to perform a differentiable architecture search (DARTS), which led to novel network models and required reasonable resources (few days using 1-4 GPUs).

NAS methods learn from labeled data. During the search process, various architectures are considered and their value is estimated based on their performance on annotated examples. However, acquiring large amounts of human-annotated data is expensive and time-consuming, while unlabeled data is much more accessible. As most NAS techniques depend on annotations availability, their performance deteriorates when the number of annotations per each class is small.

The dependency on labeled data is not unique only to NAS but is a common problem in deep learning. Large-scale annotated datasets play a critical role in the remarkable success of many deep neural networks, leading to state-of-the-art results in various computer vision tasks. Considering how expensive it is to acquire such datasets, a growing body of research is focused on relieving the need for such extensive annotation effort. One promising lead in this direction is self-supervised learning (SSL) Doersch et al. (2015); Zhang et al. (2016); Noroozi & Favaro (2016). Self-supervised methods learn visual features from unlabeled data. The unlabeled data is used to automatically generate pseudo labels for a pretext task. In the course of training to solve the pretext task, the network learns visual features that can be transferred to solving other tasks with little to no labeled data. Contrastive SSL is a subclass of SSL that has recently gained attention thanks

to its promising results Chen et al. (2020a); He et al. (2020); Chen et al. (2020b); Tian et al. (2019); Oord et al. (2018). This family of techniques contrasts positive samples and negative samples to learn visual representations.

**Contribution.** Inspired by the success of SSL for learning good visual representations, we apply an advanced self-supervised learning technique in NAS to rectify its limitation with respect to the availability of data annotations. Our Self-Supervised Neural Architecture Search (SSNAS) framework can find novel architectures without relying on data annotations. Instead of using self-supervision to learn visual representations, we employ it to learn the architecture of deep networks (see Figure 1). A recent work Liu et al. (2020) also have shown that unsupervised neural architecture search achieves results comparable to those of the supervised benchmarks. However, their experimental setting is different. Their main focus is on showing that SSL can be used to perfrom NAS with no labels. In our work we further support their claim by using a more advanced SSL technique, namely, SimCLR. More importantly, while they focus only on showing that labels are not necessary for NAS, we address an additional question: can the architectures learned without annotations improve self-supervised representation learning? We demonstrate for the case of limited resources that using an architecture found by NAS (trained without labels) improves SSL performance. This is a contribution that is unique to our work.

We apply our strategy with the popular DARTS Liu et al. (2019b) method. We adopt their differentiable search, which allows using gradient-based optimization, but replace their supervised learning objective with a contrastive loss that requires no labels to guide the search. In particular, we adopt the method used in the SimCLR framework Chen et al. (2020a). This approach for learning visual representations has recently achieved impressive performance in image classification. We adapt their approach to the architecture search process. We perform a composition of transformations on the inputs, which generates augmented images and look for the model that maximizes the similarity between the representations of the augmented images that originate from the same input image. As the focus of this work is on efficient search, we limit the used batch sizes to be the ones that can fit a conventional GPU memory. This allows SSNAS to efficiently learn novel network models without using any labeled data.

We demonstrate that our self-supervised approach for NAS achieves results comparable to the ones of its equivalent supervised approach. While these findings are similar to Yan et al. (2020); Liu et al. (2020) but with a different strategy, we further show that SSNAS does not only achieves the same results as supervised NAS, but it also succeeds in some scenarios where the supervised method struggles. Specifically, SSNAS can learn good architectures from data with a small number of annotated examples available for each class. We also demonstrate the potential of using NAS to improve unsupervised learning. We show some examples where SSL applied with the learned architectures generates visual representations that lead to improved performance. We claim that in addition to the fact that NAS can benefit from SSL, also SSL can benefit from NAS.

## 2 Related work

**Neural architecture search.** The first methods to perform neural architecture search focused on using reinforcement learning Zoph & V. Le (2017) and evolutionary (genetic) algorithms Real et al. (2017; 2018). They have shown that the architecture found using these techniques outperform the performance achieved by manually designed models. The disadvantage of these approaches is their very long search time and the need for a significant amount of resources.

To overcome the computational issue, efficient searches have been proposed. These include the effective NAS (ENAS) Pham et al. (2018) and differentiable architecture search (DARTS) Liu et al. (2019b). The first reduces the computational load of RL models using a graph structure and the second model searches in a differentiable manner, which makes it computationally efficient. These approaches have been further extended to improve the search Cai et al. (2019); Noy et al. (2020); Liang et al. (2019); Chen et al. (2019); Xie et al. (2019b); Tan et al. (2019); Wu et al. (2019) and also for applying it to applications beyond classification such as semantic segmentation Liu et al. (2019a), medical image segmentation Weng et al. (2019); Zhu et al. (2019), object detection Ning et al. (2020), image generation Wang & Huan (2019), few-shot learning Doveh et al. (2019), etc.

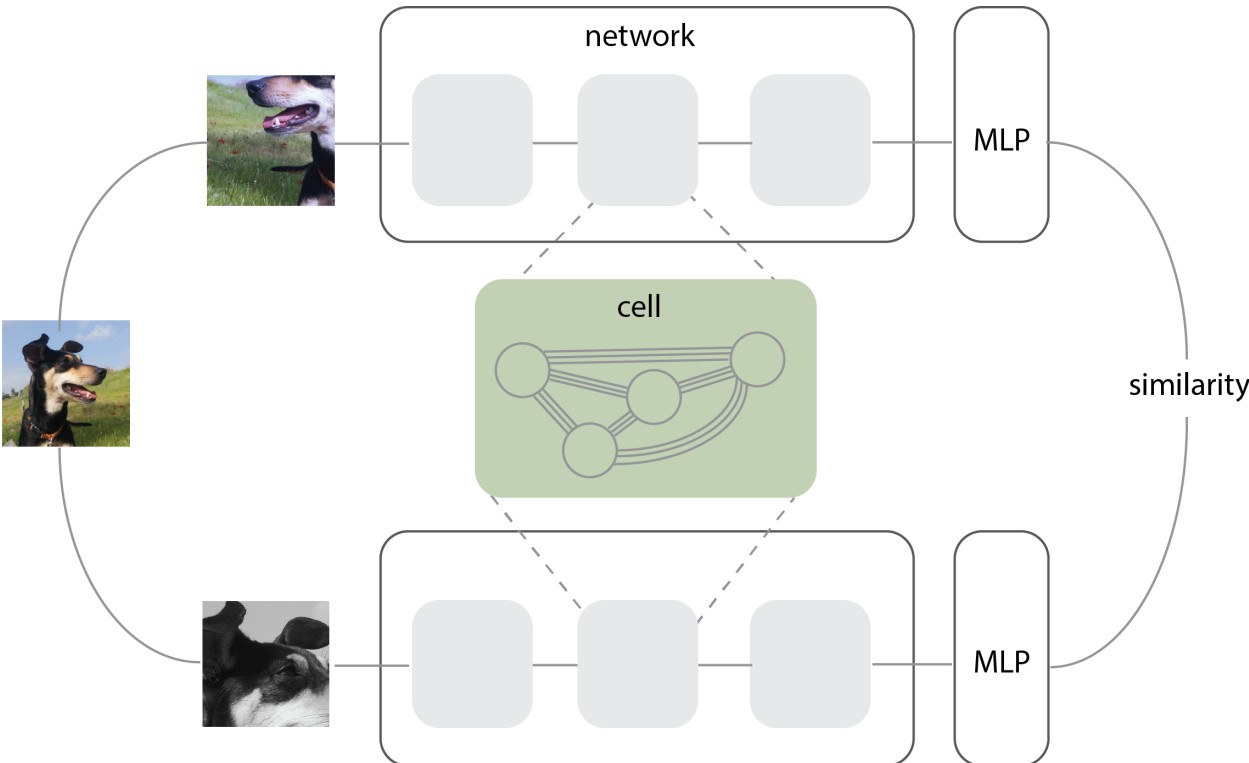

Figure 1: **The SSNAS framework.** We perform network architecture search in an unsupervised manner (with no data labels) by using a contrastive loss that enforces similarity between two different augmentations of the same input image. Both augmentations pass through the same network (i.e., the weights are shared).

One of the disadvantages of most search methods is that they require labeled data for the search. Moreover, when the number of training examples per class in a given dataset is low, the search becomes less stable. For example, while DARTS gets very good network architectures on CIFAR-10 Krizhevsky (2009) that has 5000 labeled examples per class, its performance degrades significantly on CIFAR-100 Krizhevsky (2009), where each class contains only 500 labeled examples per class. Follow-up works Liang et al. (2019); Chen et al. (2019) have added a regularization on the searched operations to mitigate these issues. Yet, an important question is whether such regularizations that require prior knowledge on the target network structure are needed.

In this work, we use a technique to perform a search on the data without the labels at all, which leads to a stable search using even vanilla DARTS. Moreover, it allows performing the search on datasets with no (or only a few) labels where none of the above methods is applicable.

**Self-supervised learning.** A considerable amount of literature was published on self-supervised representation learning. These studies suggested various pretext tasks where pseudo labels generated from unlabeled datasets drive networks to learn visual features. There is a wide choice of such pretext tasks available in the literature: predicting patch position Doersch et al. (2015), image colorization Zhang et al. (2016), jigsaw puzzles Noroozi & Favaro (2016), image inpainting Pathak et al. (2016), predicting image rotations Gidaris et al. (2018), etc. Using this approach, researchers achieved good results. However, the generality of the representations produced is arguably limited due to the specific nature of the pretext tasks.

Recent attention was given to the contrastive self-supervised learning Chen et al. (2020a); He et al. (2020); Chen et al. (2020b); Tian et al. (2019); Oord et al. (2018). Methods that use this concept such as SimCLR Chen et al. (2020a) and MoCo He et al. (2020) attained promising results, narrowing significantly the gap between supervised and unsupervised learning. SimCLR Chen et al. (2020a) employed a composition of data

augmentations on input images to create different views of the same image and used a nonlinear head on the representation before applying a contrastive loss. MoCo He et al. (2020) maintained a dynamic dictionary and aimed at maximizing the similarity between an encoded query and keys encoded by a slowly progressing momentum encoder. The follow-up work, MoCo v2 Chen et al. (2020b) adopted a few techniques from SimCLR to further improve MoCo performance.

A previous study Kolesnikov et al. (2019) on self-supervised representation learning has established that model design choices that lead to improved performance for supervised learning, are not necessarily the optimal design choices for self-supervised learning. Six models have been investigated and design choices such as the width of the model and the number of filters have been analyzed. Our work is complementary to that work since we also consider different models for SSL, only that we suggest employing a complete self-supervised architecture search, using a gradient-based optimization to discover superior architectures (as opposed to only considering a few manually designed architectures, which gained popularity based on their performance in the supervised setting).

A recent work Liu et al. (2020) shows that architectures learned without using labels are comparable in performance to architectures learned by supervised methods. The authors experiment with SSL methods different than ours (namely rotation prediction, colorization, and solving jigsaw puzzles). They investigate whether NAS can be executed without using labels, while we also demonstrate the improved performance of self-supervised representation learning using the architectures learned by self-supervised NAS in the case of limited resources. Note that the work in Liu et al. (2020) shows only the impact of SSL for NAS and not of NAS for SSL as demonstrated in this work.

Another study Yan et al. (2020) separates architecture representation learning from architecture search and shows how an unsupervised version of the first can help the latter in terms of efficiency and robustness. Though architecture embeddings are learned in an unsupervised manner, their search still requires labeled data, while our self-supervised search does not require any labels. They also adopt a very different strategy than ours (namely, they use Reinforcement Learning and Bayesian Optimization) and introduce unsupervised learning through Variational Graph Isomorphism Autoencoders while we utilize contrastive self-supervised learning. As mentioned above, we also show the advantage of NAS for SSL, which is not discussed in Yan et al. (2020).

## 3 Method

Our SSNAS framework aims at finding effective models without using data annotations. Instead, it learns by maximizing the similarity between projected representations of different views originating from the same input image. We focus on the case of limited resources and opt for approaches we can carry out using a single GPU.

Our approach is inspired by the following key observation: Virtually, all currently used networks overfit the training data, i.e., if we compare the randomly sampled networks to the found ones, all of them attain close to zero training error. Therefore, the difference between them is their generalization ability. In other words, the search goal is to find the training architecture that inherently generalizes best the data.

Given the above, an important question is whether we can find networks that generalize well without using the labels? To do so, we first revisit the NAS problem and present it from the perspective of learning to generalize well on the training data. Then we discuss how one may improve generalization without access to the labels. Namely, we rely on recent theoretical findings that show that networks with a large margin exhibit good generalization abilities Xu & Mannor (2012); Liu et al. (2016); Sokolić et al. (2017); Bartlett et al. (2017); Neyshabur et al. (2018).

With this perspective in mind, we suggest using a recent self-supervised technique, SimCLR, which uses the contrastive loss that increases the network margin in an unsupervised way, to perform an unsupervised architecture search (we demonstrate our approach on the DARTS strategy). Specifically, we train the network on one part of the data to increase the margin between examples and select the network architecture that succeeds to maintain the largest distance between samples that were not present during training (i.e., the

one that can achieve the largest margin). We start by presenting our general framework and then briefly describe the used SimCLR and DARTS approaches.

### 3.1 Margin based search

In essence, any architecture search technique splits the data into two parts, the train set $\mathcal{T}_T$ and the validation set $\mathcal{T}_V$, and then tries to find the best architecture $f_W$ (with a set of weights $W$) from a certain search space $\mathcal{A}$ that leads to the best performance on the validation set, i.e., it aims at solving an optimization problem of the form

$$\min_{f_{W^*} \in \mathcal{A}} E(f_{W^*}, \mathcal{T}_V) \tag{1}$$
$$s.t. \quad W^* = \operatorname{argmin}_W E(f_W, \mathcal{T}_T),$$

where $E(f_W, \mathcal{T})$ is the error of the network $f_W$ on the dataset $\mathcal{T}$. Next, assume that all the networks in the search space are capable of attaining an error close to zero, i.e.,

$$\min_W E(f_W, \mathcal{T}_T) \leq \epsilon, \quad \forall f_W \in \mathcal{A}. \tag{2}$$

Under this assumption, we may rewrite Eq. (1) as

$$\min_{f_{W^*} \in \mathcal{A}} E(f_{W^*}, \mathcal{T}_V) - E(f_{W^*}, \mathcal{T}_T) \tag{3}$$
$$s.t. \quad W^* = \operatorname{argmin}_W E(f_W, \mathcal{T}_T).$$

The term $E(f_{W^*}, \mathcal{T}_V) - E(f_{W^*}, \mathcal{T}_T)$ is known as the generalization error of the network (to be precise, we need to replace $E(f_{W^*}, \mathcal{T}_V)$ with the expected error over all the data; yet the validation error is often considered to be a good proxy of the latter). Clearly, the error in the optimal architecture found by solving Eq. (3) is the same as the one found by solving Eq. (1) up to an $\epsilon$ difference.

The above discussion suggests that instead of minimizing the error in the search (as in Eq. (1)), one may aim at finding the model that is capable of attaining the smallest generalization error (as in Eq. 3). Performing a search using the latter still requires having the data labels. Yet, in the literature, various measures have been developed to upper bound the generalization error of the network Neyshabur et al. (2017); Arora et al. (2018). In this work, we focus on the margin-based approach that relates the generalization error of the network to the margin of the network Xu & Mannor (2012); Liu et al. (2016); Sokolić et al. (2017); Bartlett et al. (2017); Neyshabur et al. (2018). These works show that increasing the margin of the network, i.e., the distance from the training examples to the decision boundary of the network, improves the network's generalization error. Moreover, it is shown that if we make a network invariant to different augmentations, its generalization error improves Sokolic et al. (2017).

Therefore, we replace the labeled based search with an unsupervised search that maximizes the margin. While there are many possible directions to perform this, we focus on a self-supervised learning-based approach that optimizes the embedding space.

### 3.2 The Contrastive Loss for Self-supervised Search

To search without using labels, we adopt the SSL approach of SimCLR Chen et al. (2020a). First, a composition of random data augmentations (e.g. *crop and resize, horizontal flip, color distortion* and *gaussian blur*) is applied to input images. The augmentations of the same input are considered to be a positive pair and the augmentations of different inputs are considered to be negative pairs. The distance between views of positive pairs is minimized while the distance between negative samples is maximized.

Consider such an optimization from a margin perspective. If each input image is a class, then the optimization aims at finding the feature space with the largest margin between these classes. A network capable of finding a feature space with a large margin in this case, is also expected to find a feature space with a large margin when the classes correspond to groups of multiple input examples. (In that case, we are only concerned with

the margin between the classes, which is an easier task). This provides us with a proxy loss for NAS when not enough labels are available for the search.

In view of the above, we expect that applying NAS with a SimCLR objective will lead to finding network architectures that have good generalization properties, which is exactly our goal in the optimization in Eq. (3). While SimCLR uses a fixed ResNet He et al. (2015) to learn effective visual representation, we use our dynamic network of stacked cells with mixed operations (following the DARTS approach) to learn an effective network architecture. This choice enables us to increase the network margin by contrasting positive pairs and negative pairs generated on the fly.

Before we turn to show empirically that indeed, using this loss in NAS leads to comparable results to the supervised search, we briefly describe the DARTS approach and the SimCLR strategy in more detail. A reader that is familiar with these methods may skip directly to Section 4.

### 3.3 Differential Architecture Search

We perform neural architecture search by adapting the framework of DARTS Liu et al. (2019b). We search for a cell to be stacked to a deep network. Searching for a cell that serves as a building block for the final architecture is an efficient approach yet it comes at the expense of optimality.

A cell is represented by a directed acyclic graph (DAG) of N nodes $\{x_i\}_{i=0}^{N-1}$. Node $x_i$ represents a feature map and edge $(i, j)$ represents an operation $o_{(i,j)}$ performed on $x_i$. Cell input nodes are the outputs of two previous cells. Intermediate nodes are obtained by summing the operations performed on previous nodes: $x_j = \sum_{i<j} o_{(i,j)} x_i$. Cell output is a concatenation of all intermediate nodes.

During the search, operations are selected from the set $\mathcal{O}$, which contains the possible operations (e.g. *convolution, pooling, identity* and *zero*). To relax the search space, instead of having a specific operation $o_{i,j}$ applied to each node $x_i$, a mixture of operations is applied, i.e., we have a weighted sum of all possible operations: The candidate operations are weighted by the $\alpha_{(i,j)}$ vectors. A softmax is applied over all the weights in $\alpha_{(i,j)}$ to emphasize the ones with the larger weights. To obtain a discrete architecture once search is concluded, we select the most dominant operation by applying argmax on the $\alpha_{(i,j)}$ vectors. The set $\alpha = \{\alpha_{(i,j)}\}$ (after the pruning) encodes the architecture. To form the final network, the stacked cells are preceded by a convolutional layer and followed by a global pooling. The original DARTS also includes a linear layer (with softmax at the end), yet we replaced it with an MLP, as described in the following subsection.

The architecture $\alpha$ and the network's weights are learned jointly via solving a bilevel optimization problem. This is required since the validation loss depends on the weights that minimize the training loss, which in turn depends on alpha that is obtained by minimizing the validation loss. To solve it, the architecture gradient is computed via approximation of the weights that minimize the training loss (instead of training the network fully). According to a second-order approximation, those weights are computed by performing a single train step. The first-order approximation simply uses the current weights. Using this approximation rather than the second-order one speeds up the search, yet it comes at the cost of reduced performance. Please refer to the DARTS Liu et al. (2019b) paper for more details.

### 3.4 The SimCLR approach

In SimCLR each input image $x$, is augmented twice, forming a positive pair $\widetilde{x}_i$ and $\widetilde{x}_j$. Then, the augmented views are passed through our network of stacked cells (as in the DARTS model). Denoting the output of the network final pooling layer by $h_i$, an MLP $g(\cdot)$ is used to project it to a latent space, obtaining $z_i = g(h_i)$. As SimCLR demonstrated, this mapping is effective as presumably, it allows $h_i$ to keep all the information necessary for classification, while $z_i$ can discard some of it to predict the agreement between pairs of views more accurately.

In a batch of $N$ input images, there are $2N$ augmented images. Among them, each pair of augmented views, namely $\widetilde{x}_i$ and $\widetilde{x}_j$, form a positive pair, while the other pairs serve as negative examples. For each positive

pair, we use the normalized temperature-scaled cross entropy loss Chen et al. (2020a):

$$\ell_{i,j} = -\log \frac{\exp(\mathrm{sim}((z_i, z_j)/\tau)}{\sum_{k=1}^{2N} \mathbb{1}_{[k \neq i]} \exp(\mathrm{sim}(z_i, z_k)/\tau)},$$

where $\mathrm{sim}(z_i, z_j) = \frac{z_i^\top z_j}{\|z_i\|\|z_j\|}$ is the cosine similarity, $\mathbb{1}$ is an indicator function and $\tau$ is a temperature hyperparameter.

### 3.5   Self-Supervised Neural Architecture Search

To find network architectures that generalize best the data without accessing the labels, we have to tailor a single framework, namely SSNAS, which puts together concepts from the areas of NAS and SSL. Adapting the work of DARTSLiu et al. (2019b), we have to suggest different techniques for learning from the data without any annotations and define an objective to drive the search accordingly. In order to enhance the generalization ability of the learned models, we look for a method to increase the margin of the network. Inspired by how SimCLRChen et al. (2020a) uses contrastive loss for visual representation learning, we also use the contrastive loss and perform the strong data augmentations that encourage effective learning of the network architecture. While SimCLR uses a fixed ResNetHe et al. (2015) to learn effective visual representation, we use our dynamic network of stacked cells with mixed operations to learn an effective network architecture. This choice enables us to increase the network margin by contrasting positive pairs and negative pairs generated on the fly. In order to generalize well on the training data, we keep the procedural choice of separating the original training set into two separate sets, where one is used to learn the network weights while the other pushes towards a network structure with an increased margin.

Though SSL frameworks achieve state-of-the-art results by using many computation resources (namely Sim-CLR uses large batch sizes up to 8192 and up to 128 cores of TPUs), we focus on the case of limited resources. We use small batch sizes and work with models that fit in a single GPU.

## 4   Experiments

We turn to test our SSNAS approach and conduct a self-supervised architecture search to identify novel architectures without using labeled data. To evaluate the learned cells, we measure the performance of the found architectures using labeled data. To further examine our approach, we experiment with self-supervised pretraining[1] in the case of limited resources and evaluate the learned representations for classification with limited annotations. We show that using the learned architectures with SSL can improve the learned representations.

**Datasets.** We conduct most of our search experiments on CIFAR-10 Krizhevsky (2009). In this case, we show that search with and without labels lead virtually to the same performance. To evaluate cell transferability, we train learned cells on ImageNet Russakovsky et al. (2015). Then, we turn to other datasets where the number of available labels per class is relatively small. These include CIFAR-100 Krizhevsky (2009), where the vanilla DARTS Liu et al. (2019b) struggles, and STL-10 Coates et al. (2011), where the number of labels is significantly small and thus applying supervised NAS techniques is very challenging. We then train a classifier for the latter based on representations learned by SSL using the unlabeled data in STL showing the potential of the learned architecture to contribute to visual representation learning.

### 4.1   Implementation details

We describe now the setup we use for the architecture search with the SSL loss, both in the training and evaluation phases. We also detail the SSL pretraining of the found architectures.

**Architecture search.** We use the same setup that was detailed in DARTS Liu et al. (2019b). We learn a normal cell and a reduction cell, each consisting of 7 nodes. The candidate operations include separable convolutions and dilated separable convolutions (3x3, 5x5), average pooling (3x3), max pooling (3x3), zero,

---

[1]Here and elsewhere in the paper pretraining refers to unsupervised pretraining by SSL.

| Architecture | Search Type | Test Error (%) |
|---|---|---|
| Random sampling | random | 3.29 |
| DARTS (first order) | supervised | 3.00 |
| DARTS (second order) | supervised | **2.62** |
| SSNAS (first order) | self-supervised | **2.61** |

Table 1: Image classification test error on CIFAR-10

| Architecture | Search Type | Test Error (%) | |
|---|---|---|---|
| | | top-1 | top-5 |
| DARTS (second order) | supervised | 28.92 | 10.24 |
| SSNAS (first order) | self-supervised | **27.75** | **9.55** |

Table 2: Image classification test errors on ImageNet

| Architecture | Search Type | Test Accuracy (%) |
|---|---|---|
| DARTS (first order) | supervised | 64.46 |
| Random sampling | random | 82.61 |
| SSNAS (first order) | self-supervised | **83.36** |

Table 3: Image classification accuracies on CIFAR-100

| Architecture | Batch Size | Test Accuracy (%) | |
|---|---|---|---|
| | | Linear Evaluation | Semi-supervised |
| ResNet-18 | 32 | 87.63 | 88.85 |
| Random sampling | 32 | 83.48 | 88.39 |
| SSNAS (first order) | 32 | **88.78** | **89.45** |
| ResNet-18 | 64 | 90.53 | 90.17 |
| Random sampling | 64 | 88.13 | 90.03 |
| SSNAS (first order) | 64 | **90.87** | **90.94** |

Table 4: Learning visual representations from CIFAR-10 (limited-resources scenario)

and identity. To obtain the final cell after the search concludes, we keep for each node the two strongest operations (among all the operations from the predecessor nodes). By stacking the cells, we form a deep network. As search results might be sensitive to initialization, we run the search four times with different random seeds. In order to generalize well on the training data, we keep the procedural choice of separating the original training set into two separate sets, where one is used to learn the network weights while the other pushes towards a network structure with an increased margin.

To solve the optimization problem, we use the first-order approximation of the gradient which requires fewer resources. Even though it comes at the cost of reduced performance, we were still able to use it to get performance comparable to DARTS and even better in some cases.

Though SSL frameworks achieve state-of-the-art results by using many computation resources (namely Sim-CLR uses large batch sizes up to 8192 and up to 128 cores of TPUs), we focus on the case of limited resources: We use small batch sizes and work with models that fit in a single GPU.

**Architecture evaluation.** To select which model to evaluate, we employ the following model selection strategy as in Liu et al. (2019b): We train each network for a small number of epochs (100) and pick the best model based on its performance on a validation set. To evaluate the final architecture, we train the network from scratch and test it on a test set. The network used for model selection and training is larger than the one used for search (8 cells for search and 20 cells for training), and also the number of input channels is higher (16 channels for search and 36 channels for training).

**Self-supervised pretraining.** For pretraining, we adapt the procedure used in SimCLR Chen et al. (2020a). We use our learned architecture as the base network, and add a nonlinear head on top of it (namely an MLP with two layers and a ReLU nonlinearity). The representations are mapped to 128-dimensional projections. The composition of random augmentations includes random crop and resize, random horizontal flip, color distortion, and Gaussian blur. As we investigate the case of limited resources, we use small batch sizes of 32 and 64 on a single GPU (unlike the original SimCLR Chen et al. (2020a) framework that experimented with batches of size up to 8192 and used up to 128 TPU cores). The same settings are used also in SSNAS (to enforce the contrastive loss and perform the search without labels).

**Evaluation of the self-supervised learned representations.** We evaluate the learned representations using the common linear evaluation protocol Zhang et al. (2016); Chen et al. (2020a); Oord et al. (2018). We freeze the pretrained network and add a linear classifier on top of it, and train it on the entire train set.

We also evaluate the learned representation in a semi-supervised setting. We sample 10% of the labels in a given labeled dataset and fine-tune the entire pretrained network on the labeled data. We conclude with testing the fine-tuned model on the test set. For datasets that are already suitable for semi-supervised learning (e.g., STL), we use the few provided examples (instead of taking a fraction of the annotations). These experiments show the potential of the learned architecture to contribute to visual representation learning with SSL.

### 4.2 Learning network architectures from unlabeled data

**Comparisons to the fully annotated case.** We used our SSNAS framework to search for novel architectures on CIFAR-10 without using any annotations. For model selection, we ran a short self-supervised train (100 epochs) for each of the learned architectures and measured its performance on the validation set based on the model's contrastive loss (without using annotations). For evaluation purposes, we then performed a full supervised train (650 epochs) on the selected model and tested it on the test set.

Table 1 compares our results to the baseline (DARTS) that requires data annotations. Notice that our search, which uses only a first-order approximation, is comparable to the results of DARTS with the second-order approximation and outperforms both the first order DARTS and the random sampling. Remarkably, this is obtained without using any labels during the search.

We investigated cell transferability by evaluating the learned model on ImageNet Russakovsky et al. (2015). We adjusted the model to have 14 cells and 48 input channels. We trained the network from scratch for 250 epochs and tested its performance on the test set. We used the same hyperparameters as in DARTS

| Architecture | Fine-tune | Test Acc. (%) |
|---|---|---|
| Random sampling | pretrained model | 84.55 |
| ResNet-18 | pretrained model | 86.13 |
| SSNAS (first order) | random weights | 67.00 |
| SSNAS (first order) | pretrained model | **86.70** |

Table 5: Learning visual representations from STL-10 (semi-supervised setting)

Liu et al. (2019b) except for a bigger batch size to speed up the training. The same settings were used to evaluate SSNAS and DARTS. Table 2 shows SSNAS results for cell transferability against the results of DARTS. These results confirm that SSNAS matches the performance of the supervised approach also when transferring the architecture.

**Comparisons to scarce labels case.** To experiment on datasets for which the number of labeled examples per class is relatively small, we also employed SSNAS to search for architectures on CIFAR-100 (following the same procedure described for CIFAR-10). Table 3 shows comparisons between SSNAS results and the results of the baseline. In this case, we show that our method succeeds while the vanilla DARTS struggles, without adding regularization or employing specific techniques to prevent DARTS collapse. We also show that SSNAS outperforms random sampling as well. This experiment demonstrates the advantage of performing architecture search with no labels when the number of labeled examples per class is relatively small.

### 4.3 Self-supervised learning using searched models

To investigate the potential of using NAS to improve unsupervised learning, we used the architectures we found for CIFAR-10 as the base network for learning visual representations from CIFAR-10 using SSL in the case of limited annotations. We carried out the model selection by running short pretraining on each of the searched architectures and then selected the best performing network based on the model's contrastive loss on the validation set (without using any annotations). We then pretrained the selected model with a relatively small batch size (32 or 64) as we consider the limited-resources scenario. To evaluate our results, we applied SSL also with randomly selected architectures and with ResNet-18. We compare to ResNet-18 as this is the architecture employed in Chen et al. (2020a) for CIFAR-10. For evaluation, we used the common linear evaluation protocol and the semi-supervised settings.

Table 4 presents our results and compare them to the randomly selected architectures and SimCLR's base network (ResNet-18). Notice that the learned architecture attains better performance for the batch sizes that are considered compared to both the randomly selected architecture and the ResNet-18 model. This shows the potential of combining SSL with NAS to learn improved visual representations.

To assess the applicability of our framework on datasets that are suitable for semi-supervised learning with only a few labels available, we experimented on STL-10 Coates et al. (2011). As the number of annotations is relatively small (500 labeled training examples per class), we cannot apply supervised methods for the search. Instead, we used SSNAS to search for a compact architecture (with 5 cells) that served as the base network for pretraining. In the next step, we used the labeled examples in the training set (5000 training images) to finetune the network. We repeated this procedure several times, with the following changes: (i) using a random architecture instead of the learned one; and (ii) training from scratch (initializing the architecture with random weights) rather than using the weights of the pretrained model. For comparison, we also performed pretraining with ResNet-18He et al. (2015) as the base model using the same settings. Results of this experiment are presented in Table 5. Fine-tuning the pretrained model learned by SSNAS outperforms fine-tuning the original ResNet-18 model, fine-tuning a random model or using the learned model without the learned weights. This experiment demonstrates the advantage of pretraining with a learned architecture on datasets that are suitable for semi-supervised learning and confirms the potential of NAS to improve SSL.

### 4.4 The train set and the validation set of SSNAS

During the search, we split the data into two parts, a train set used to learn the network weights, and a validation set used to learn the architecture parameter $\alpha$. An interesting question is whether to perform such a split or to use the same set (containing all the samples) for both learning the network's weights and learning $\alpha$. While for supervised search the first is practiced, we wanted to verify that this is also the preferred choice for our SSL search. Our experiments have shown that indeed using separate sets leads to improved performance compared to using the same set (with twice as many examples): We observe a difference in performance on the test set of 0.63% in favor of splitting the data.

## 5 Conclusion

In this paper, we present a framework for self-supervised neural architecture search. This study set out to support the claim that architecture search can be carried out without using labeled data. We demonstrated this with the SimCLR approach, which is a leading technique in SSL. Indeed, our framework matched the performance of equivalent supervised methods without using annotations at all.

Our work also identified that the learned architectures can be used as the base network in SSL frameworks and improve their performance. In particular, we exhibited this advantage for datasets with few labels, i.e., using SSL in the limited annotations scenario. Our findings demonstrated that SSL and NAS can be put in a symbiosis where both benefit from each other.

The focus of this work is exploring the search and training with limited resources. A natural progression of this work is to expand the experiments on SSL to larger models, datasets, and batch sizes. One may also experiment with other recent learning methods such as self-training with noisy student Xie et al. (2019a). We believe (although we do not have the resources to check that) that the same advantage will be demonstrated in these cases. Notwithstanding these limitations, the results established in this work already demonstrate the great advantage of using SSL and NAS together.

Another possible follow-up research direction is using an architecture search to learn better augmentations to be used with the self-supervised learning techniques. This can be done for example by extending methods such as auto-augment that searches for the optimal augmentations for a given supervised task Cubuk et al. (2018); Lim et al. (2019). While Chen et al. (2020a) reported that using the current augmentations found by auto-augment does not improve the learned representations in their method, we believe that by combining the two such that the learned augmentations are designed specifically for the SSL task, the learned visual representations can be improved.

Finally, in this work, we also established a margin view of SSL when used as a loss for architecture search. We believe that this view has the potential to further be developed and perhaps lead to an explanation for the general SSL success.

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
