# OpenReview forum: "Margin based Self-Supervised  Neural Architecture Search"
_TMLR — Rejected by TMLR_

### Review · Reviewer_mvnD · 2022-08-15

**Summary Of Contributions:**

This paper combines the NAS method with a self-supervised learning scheme. For neural architecture search, this paper adopts a classic gradient-based method (DARTS). The widely-used self-supervised loss（SimCLR）is used to replace the common cross-entropy loss in supervised learning. It validates that NAS and SSL learning can be well combined.

**Broader Impact Concerns:**

No concerns on the ethical implications。

**Requested Changes:**

Please see the weaknesses.

**Strengths And Weaknesses:**

Strength:

-Combing NAS and self-supervised learning is reasonable to achieve high performance.

-The proposed method is simple and easy to follow.

Weakness:

-This paper directly combines two methods and brings limited new insight. DARTS is a classical gradient-based NAS method and SimCLR is a widely-used SSL loss. These methods work well in their respective fields and it is not surprising that combing them can achieve good performance.

-Eqs.(1)-(3) have a weak relationship with the proposed method. Eqs.(1)-(3) are theoretical formulations, which aim to find architectures with low generalization errors. While this paper adopts  an existing self-supervised strategy (i.e.g, SimCLR) directly and claims it can reduce the gaps between training errors and testing errors. The authors do not design methods or analyze the problem theoretically based on Eqs.(1)-(3). Thus these formulations are redundant.

-For the experimental parts, the authors only compare the proposed with DARTS, which is a classical method. Many methods proposed recently have achieved much higher performance than DARTS, such as [1,2,3]. It is required to compare with them to investigate the superiority of SSNAS.

-In all the tables (such as Table 1,2,3), only the test errors are reported. Actually, there are other important properties to judge a NAS method. For example, the computational cost, latency, and number of parameters are important properties for an architecture. The search cost is also a vital indicator to validate whether the proposed method is efficient.


[1] Fair DARTS: Eliminating Unfair Advantages in Differentiable Architecture Search

[2] Fast Neural Network Adaptation via Parameter Remapping and Architecture Search

[3] Rethinking Performance Estimation in Neural Architecture Search

---

### Review · Reviewer_3qe2 · 2022-09-08

**Summary Of Contributions:**

This paper proposes a self-supervised neural architecture search (SSNAS) method that allows finding novel network models without the
need for labeled data. SSNAS achieves comparable results with supervised NAS, and it can improve the performance of self-supervised learning. Empirical results on ImageNet, CIFAR and STL-10 are presented.

**Requested Changes:**

I think the authors may consider improving the contributions of this work on top of [*1]. For example, would it be possible to perform NAS on top of the masked autoencoder (i.e., MAE)?

**Strengths And Weaknesses:**

Strengths:

- This paper discusses the similarities between SSNAS and [*1] honestly. I highly appreciate this point.

Weaknesses:

- However, I have to say that, I do not see many contributions of this work on top of [*1]. The authors propose to use simCLR as the unsupervised learning algorithm, and demonstrate that their searched architecture is more effective in terms of self-supervised learning. As a matter of fact, both the two extensions seem to be trivial and incremental.


[*1] Chenxi Liu, Piotr Dollár, Kaiming He, Ross Girshick, Alan Yuille, and Saining Xie. Are labels necessary for
neural architecture search? In ECCV, 2020.

---

### Review · Reviewer_7PFC · 2022-09-13

**Summary Of Contributions:**

The authors investigate the problem of integrating self-supervised learning (SSL) into neural architecture search (NAS). A self-supervised neural architecture search (SSNAS) framework is proposed, which leverages unlabeled data and is expected to find network with strong generalization ability. SSNAS is based on a differentiable NAS method DARTS, and adopts SimCLR as the SSL method. The authors show that their SSNAS well matches the performance of supervised NAS, and further reveal that SSL can also benefit from NAS.

**Broader Impact Concerns:**

I have no broader impact concerns.

**Requested Changes:**

See my comments in Weaknesses.

**Strengths And Weaknesses:**

Strengths:
1. Experiments show that the proposed SSNAS with only the first-order approximation outperforms its supervised version DARTS with second-order approximation, validating the effectiveness of SSNAS.
2. The authors not only validate the benefit of performing NAS in a self-supervised way but also show that NAS can improve the performance of SSL.
3. Unlike classic contrastive learning methods, the proposed SSNAS does not require a large batch size, which is resource-saving.

Weaknesses:
1. The proposed method is not novel, as the NAS baseline method DARTS and the SSL method SimCLR are not new.
2. To make the experiments more convincing, different NAS/SSL methods should also be experimented.
3. The authors pay too much attention on the preliminary in Section 3 instead of their proposed method.

---

### Review · Reviewer_K9CR · 2022-09-13

**Summary Of Contributions:**

This paper proposes Self-Supervised-learning (SSL)-based Neural Architecture Search (SSNAS), which aims to search for optimal neural architecture without labeled dataset during search process. To validate this approach, this work pick DARTS which is one of representative NAS methods and replace the existing supervised learning objective of DARTS into a contrastive loss which do not need labeled data inspired by SimCLR. The experimental results of the proposed method are competitive with the results of supervised NAS works. Interestingly, this work shows that the architecture learned without labels can improve the performance of the self-supervised representation learning, in other words, they reveal that SSL can benefit from NAS.

**Broader Impact Concerns:**

There is no concerns and the ethical implications of the work.

**Requested Changes:**

Please see the comments in Weaknesses part.

**Strengths And Weaknesses:**

- Strengths
1. SSNAS with the first order and using not annotated dataset outperformed the baseline DARTS with the second order and using fully annotated dataset.
2. This work applied SimCLR which is one of the representative Self-supervised learning methods to NAS and validated the effectiveness of it.
3. This work showed the benefit of NAS to the SSL.

- Weaknesses
1. This work is not novel. As the authors mentioned, there are several existing self-supervised-learning-based NAS methods. The reviewer thinks different data augmentation or applying different self-supervised method are minor contributions.
2. Weak baselines. This work did not compared with other self-supervised learning-based NAS methods. DARTS and ResNet-18 are rather native and old methods.
3. Experimental setting is unclear. This work only reported the test errors. However, search time or the size of architectures are one of important metric in NAS domain. The reviewer believe that authors also need to compare such things in the experiment.

---

### Author Response · Authors · 2022-10-04
**Answer to reviewers**

We would like to thank the reviewers for taking the time to review the work.
We will take them into account and will strengthen the theoretical part of the work as we do not have the computational resources to make the extra experiments proposed. Making the ones presented in the paper was already very prohibitive given our resources. Yet, we believe that showing that SSL benefits from a search is novel and was not shown before our work.

---

### Decision · Action_Editors · 2022-10-20

**Recommendation:** Reject

**Comment:**

This paper proposes a NAS method under the setting of self-supervised learning. The reviewers raise various concerns about the novelty (the NAS baseline method DARTS and the SSL method SimCLR are not new), contribution (it is a minor contribution to apply the self-supervised method directly), and experiments (more methods and metrics should be compared). The author’s response does not resolve these concerns and all the reviewers give negative recommendations. Therefore, this paper is not ready to be published on TMLR.

**Audience:**

Few individuals will be interneted in this paper as most of the findings are not new.

**Claims And Evidence:**

No. Some important metrics such as search time or architecture size are missing. The comparison with competing methods is not sufficient.